# Identification of Phage Receptor-Binding Protein Sequences with Hidden Markov Models and an Extreme Gradient Boosting Classifier

**DOI:** 10.3390/v14061329

**Published:** 2022-06-17

**Authors:** Dimitri Boeckaerts, Michiel Stock, Bernard De Baets, Yves Briers

**Affiliations:** 1Laboratory of Applied Biotechnology, Department of Biotechnology, Ghent University, 9000 Ghent, Belgium; dimitri.boeckaerts@ugent.be; 2Research Unit Knowledge-Based Systems (KERMIT), Department of Data Analysis and Mathematical Modelling, Ghent University, 9000 Ghent, Belgium; michiel.stock@ugent.be (M.S.); bernard.debaets@ugent.be (B.D.B.); 3Lab of Bioinformatics and Computational Genomics (BIOBIX), Department of Data Analysis and Mathematical Modelling, Ghent University, 9000 Ghent, Belgium

**Keywords:** phage, receptor-binding protein, hidden Markov models, machine learning, extreme gradient boosting

## Abstract

Receptor-binding proteins (RBPs) of bacteriophages initiate the infection of their corresponding bacterial host and act as the primary determinant for host specificity. The ever-increasing amount of sequence data enables the development of predictive models for the automated identification of RBP sequences. However, the development of such models is challenged by the inconsistent or missing annotation of many phage proteins. Recently developed tools have started to bridge this gap but are not specifically focused on RBP sequences, for which many different annotations are available. We have developed two parallel approaches to alleviate the complex identification of RBP sequences in phage genomic data. The first combines known RBP-related hidden Markov models (HMMs) from the Pfam database with custom-built HMMs to identify phage RBPs based on protein domains. The second approach consists of training an extreme gradient boosting classifier that can accurately discriminate between RBPs and other phage proteins. We explained how these complementary approaches can reinforce each other in identifying RBP sequences. In addition, we benchmarked our methods against the recently developed PhANNs tool. Our best performing model reached a precision-recall area-under-the-curve of 93.8% and outperformed PhANNs on an independent test set, reaching an F1-score of 84.0% compared to 69.8%.

## 1. Introduction

Antimicrobial resistance is increasingly becoming a threat to human health across the world [1]. Bacteriophages (phages for short) are considered as an alternative treatment against multidrug-resistant bacteria [2,3]. As natural predators of bacteria, they often have a narrow host specificity at the strain level [4]. For various phage applications, this can urge the discovery of new phages in nature (a so-called phage hunt), which is a labor- and time-intensive endeavor [5,6]. However, recent progress in synthetic biology has enabled the precise engineering of phages and their specificity towards bacterial hosts [7]. More specifically, the modification or swapping of receptor-binding proteins (RBPs) between phages allows to adjust the host specificity, avoiding the need to discover and cultivate new phages [8,9,10]. RBPs are key determinants of host specificity and encompass tail fibers, tailspikes, and tail tips [11]. Despite the enormous potential, it remains a challenging task to modify RBPs without losing infectivity, because of their multiple interactions with both host receptors and structural phage tail proteins [7].

Today, ever-increasing quantities of omics data together with the computational tools to analyze such data present various opportunities for novel insights in the phage field [12,13]. More specifically, the widespread availability of publicly available phage genome sequence data can further enhance our understanding of RBPs, which would, in turn, translate to more successful phage engineering efforts. Tools for the automated identification and annotation of phage proteins (including RBPs) using predictive models are good examples of this. However, the construction of such models is hampered by nonstandard or missing annotations for many phage proteins. Often, up to half of the genes of publicly available phage genomes have no proper annotation [14,15].

To address this shortcoming, various research groups have developed approaches to identify and annotate different phage proteins from sequence data. For example, Cantu et al. (2020) developed a novel approach to classify ten major classes of structural phage proteins using eleven 4-layer artificial neural networks [14]. Furthermore, Li et al. (2020) employed machine-learning models to predict phage enzymes and hydrolases, while Fernández-Ruiz et al. (2018) collected profile hidden Markov models (HMMs) to identify novel endolysins in uncultured phage genomes [16,17]. Interestingly, Cantu et al. showed that tail fiber sequences are particularly difficult to classify correctly [14]. Regrettably, because tail fibers are among the most important proteins for the infection process and are of interest for phage engineering purposes. For this reason, we aimed to improve the complex identification and annotation of RBP sequences in publicly available phage genome data using two parallel data-driven approaches.

Both strategies involve a shared, comprehensive processing pipeline to identify annotated RBP sequences in phage genomic data based on the variety of keywords that are typically used to refer to RBPs. Our first approach was inspired by the strong evolutionary pressure on RBPs and the horizontal gene transfer events occurring across RBPs [18]. As a result, RBPs are typically modular proteins consisting of a combination of N-terminal (structural) domains and C-terminal (cleaving, binding, or chaperone) domains. We, therefore, detected RBP sequences based on protein domains represented as HMMs. We manually collected HMMs that represent RBP-related protein domains from the Pfam database [19]. Subsequently, we detected RBP sequences with these domains and then built additional HMMs for the parts of detected sequences that did not correspond with a known RBP-related HMM in Pfam to further increase the potential for the discovery of RBP sequences. Our second approach involved a machine-learning-based classifier that uses state-of-the-art protein language embeddings to accurately discriminate between phage RBPs and other phage proteins [20]. In this paper, we show that both methods are complementary to some extent and explain how they can reinforce each other in identifying RBP sequences. In addition, we benchmarked our methods against the recently developed PhANNs tool [14]. Our best-performing model reached a precision-recall area-under-the-curve of 93.8% and outperformed PhANNs in terms of RBP prediction on an independent test set, reaching an F1-score of 84.0% compared to 69.8%. We released our code on GitHub (https://github.com/dimiboeckaerts/PhageRBPdetection, accessed on 10 June 2022) and Zenodo (https://doi.org/10.5281/zenodo.6491321, accessed on 10 June 2022) for the research community to use freely and expand on. In this way, we contributed to the current progress in RBP engineering efforts to adjust host specificity.

## 2. Materials and Methods

### 2.1. Phage Genome Sequence Data

A schematic overview of all consecutive data processing steps is given in Figure 1. We downloaded a total of 28,060 phage genomes via the INPHARED collection on 1 December 2021 [12]. Of those, 21,047 genomes were indicated as complete (in INPHARED) and were subjected to further processing.

We processed these phage genomes further in two steps. In the first step, we used Biopython [21] to loop over every genome record to collect all corresponding coding DNA sequences (CDSs). Each CDS of each genome (Genbank) record in the INPHARED database was checked for its annotation and assigned to one of two groups based on our collection of annotation keywords related to RBPs. These keywords included tail fiber, tail spike, receptor-binding protein, receptor-recognizing protein, and variations of these (e.g., tail fibre). A regular expression was used to detect all keywords and variations simultaneously (Figure 1, Table 1a). In the second step, sequences in both groups (denoted as *RBPs* and *Others*) were further filtered to ensure a high quality of the final datasets. Three filters were applied to the *RBP* group. First, we discarded sequences containing unknown amino acids (AAs). Second, sequences containing additional keywords not related to RBPs were discarded as well. For example, these keywords included ‘assembly’ and ‘hinge’ (among others), denoting phage proteins that are related to RBPs, but are not RBPs themselves. A full list of all discarded annotations is presented in Table 1b. Third, *RBP* sequences shorter than 200 AAs or longer than 1500 AAs were discarded, which reflects the range in length in which we expect RBPs based on Latka et al. (2019) [22]. The *Others* group was also filtered in three steps. Here, as well, sequences containing unknown AAs were discarded. Second, sequences annotated as hypothetical, putative, or uncharacterized were discarded, as well, because these sequences might be RBPs without a proper annotation. Third, *Other* sequences shorter than 30 AAs were removed. Furthermore, duplicate sequences were identified both within each group and across both groups and subsequently discarded. Finally, a subset of the *Others* group was randomly sampled with a size of 10 times the number of *RBP* sequences (6176 *RBP* sequences and 61,760 *Other* sequences in total).

As a final processing step, both datasets were split into portions for training (constructing HMMs and machine-learning models) and testing (benchmarking against PhANNs) based on their submission date. More specifically, all sequences of genome records added up until September 2021 were used for training (4189 *RBP* sequences and 37,022 *Other* sequences in total), while sequences of records that were added from October until December 2021 were kept apart for the benchmark against PhANNs (1987 *RBP* sequences and 24,738 *Other* sequences in total). This ensured that both PhANNs (trained before 2021) and our approaches had not seen any of the test sequences yet in the benchmark. Pairwise alignments of *RBP* sequences between the training set and test set show that the similarity within the training set is not different from the similarity between the training and the test set (Appendix A).

### 2.2. Collecting and Constructing Profile HMMs Related to RBPs

To identify RBPs based on protein domains, profile HMMs related to RBPs were collected from the Pfam database. More specifically, an iterative procedure was performed to discover domains related to RBPs, starting from the *Phage_T7_tail* domain, a well-known structural phage domain in the T7 phage [23]. Inspired by modularity and horizontal gene transfer events, this iterative procedure involved alternately identifying domains at either the N- or C-terminus that were linked to other RBP-related domains (at the C- or N-terminus, respectively) retrieved in the previous iteration (Figure 2). This list was further manually curated to exclude domains that were too broad (not uniquely related to RBPs) or related to other phage proteins (e.g., lysins). Reflecting the underlying biology, domains were grouped into N-terminal (structural) domains and C-terminal (cleaving, binding, or chaperone) domains. Based on previous work by Latka et al. (2019), we chose the first 200 amino acids as the cutoff for the division between N-terminus and C-terminus [22].

Subsequently, these HMMs were used to scan all the annotated RBP sequences. Sequences containing either a domain at the N-terminus or C-terminus, without any corresponding domain at the opposite end, were identified and grouped together. These sequences were used to construct new HMMs in three consecutive steps using the HMMER software [24]. First, the N-terminal parts (up until 200 amino acids) of sequences with an unknown N-terminal domain and the C-terminal parts (starting from the last known N-terminal domain) of sequences with an unknown C-terminal domain were separately clustered using CD-HIT with standard settings [25]. Second, multiple sequence alignments were constructed for each cluster containing five or more sequences using Clustal Omega with the standard settings [26]. Third, these multiple sequence alignments served as input for the *hmmbuild* functionality in HMMER to construct new HMMs from (http://hmmer.org, accessed on 10 June 2022). These custom-built HMMs were added to our set of Pfam domains as a final set of RBP-related HMMs to make detections with. This final set of collected and custom-built HMMs was made available in our GitHub repository.

### 2.3. Training an Extreme Gradient Boosting Classifier to Discriminate Phage RBPs from Other Phage Proteins

The training data of *RBPs* and *Others* were used to construct a feature representation as input to train an extreme gradient boosting (XGBoost) classifier [27]. XGBoost is a widely popular nonlinear machine-learning method that fits a collection of decision trees sequentially in which each new tree improves the performance of the current ensemble. It was chosen for its broad and off-the-shelf applicability on unstructured data. The feature representation consisted of the ProtBert-BFD protein language embeddings. Such embeddings are typically the outputs of the last hidden layer of a large deep learning language model specifically trained on an immense amount of protein sequence data [28]. The pre-trained ProtBert model used to compute embeddings was borrowed from the *bio_embeddings* package provided by Dallago et al. (2021) [20] and was computed using NVIDIA P100 GPUs.

The computed embeddings were used as input to train an XGBoost classifier to discriminate between phage *RBP* sequences and phage *Other* sequences in binary classification. We optimized two hyperparameters using the F1-score: the maximum depth of each tree and the number of estimators. Increasing the maximum depth increases the complexity of the model but makes it more likely to overfit. The number of estimators refers to the number of boosting rounds that are done. A nested 4-fold cross-validation scheme was implemented to simultaneously tune the hyperparameters (inner loop) and measure performance (outer loop). Both the F1-score and precision-recall area-under-the-curve (PR-AUC) were computed as performance metrics to evaluate the XGBoost classifier. Every step of this training and evaluation pipeline was implemented with Scikit-learn [29].

Lastly, a final XGBoost model was trained on all data with optimized hyperparameters and saved for further use during the benchmark against PhANNs.

### 2.4. Benchmarking Our Parallel Approaches against PhANNs

Both our domain-based and machine-learning-based approaches were benchmarked against the recently developed tool PhANNs, which allows predicting 10 different classes of phage proteins, including tail fiber proteins [14]. The PhANNs code and trained model were downloaded from their GitHub repository.

Each of the protein sequences (*RBPs* and *Others*) in the held-out test set was subjected to the domain-based approach, the XGBoost classifier, and the PhANNs model to classify it as either an *RBP* or an *Other* sequence. For the domain-based approach, any sequence in which a protein domain belonging to our assembled set of HMMs was recognized was designated an *RBP*. For the machine-learning approach, the output of the classifier was one or zero, indicating that the protein was predicted as an *RBP* (1) or *Other* (0). For PhANNs, scores were given for each of its ten protein classes, and the highest score was chosen as the final prediction for each protein sequence. Predictions were made for each of the sequences in the held-out test set by each of the methods. All methods were compared by computing the F1-score on the test set predictions. Finally, a Venn diagram was constructed for the RBP sequences in the test set to illustrate the concordances and discordances among the different methods.

## 3. Results

### 3.1. Phage Genome Sequence Data

Phage genome sequences were downloaded via the INPHARED collection on 1 December 2021 [12]. The CDSs of each complete genome record were collected and checked for RBP-related annotation (Figure 1). In total, 16,496 CDSs were identified as *RBP*, while 1,811,032 CDSs were classified into the *Others* group. After filtering for hypothetical proteins, undetermined amino acids, protein length, and various related but unwanted annotation keywords, 6176 *RBPs* and 228,315 *Other* sequences remained in the database.

Both datasets were split up into a part for training (constructing HMMs and machine-learning models) and testing (benchmarking against PhANNs). Entries added up until September 2021 were used for training, while entries added from October until December 2021 were kept for testing. A total of 4189 *RBP* sequences and 37,022 *Other* sequences were designated for training, while 1987 *RBP* sequences and 24,738 *Other* sequences were kept aside for testing.

### 3.2. Collecting and Constructing Profile HMMs Related to RBPs

Profile HMMs related to RBPs were manually collected from the Pfam database, iteratively starting from the well-known *Phage_T7_tail* domain (Figure 2). Thirty domains were collected and grouped into N-terminal (structural) domains and C-terminal (cleaving, binding, or chaperone) domains, reflecting their underlying biology (Table 2). These HMMs were used to scan all the annotated RBP sequences and to identify N- or C-terminal sequence parts that did not correspond to any known HMM. In total, 1786 of the 4189 annotated RBP sequences were identified as containing an RBP-related Pfam domain. Within this group of sequences, the *Phage_T7_tail*, *Tail_spike_N* and *DUF3751* domains were identified most often at the N-terminus, while the *Peptidase_S74*, *Collar* and *DUF1983* domains were most often occurring at the C-terminus (Figure 3). Furthermore, of those 1,786 sequences, a subset of 646 contained an identified N-terminal domain without an identified C-terminal domain. A subset of 874 sequences contained a known C-terminal domain without a known N-terminal domain. Of the sequences with a known N-terminal and/or C-terminal domain, 37 different architectures occurred more than five times (Figure 4). Interestingly, the five most occurring architectures were single-domain architectures, indicating that many RBP-related domains are not present in the Pfam database. For this reason, custom HMMs were constructed to increase the number of RBP detections. The N-terminal parts (up to 200 amino acids) of the 874 sequences without a known N-terminal domain were grouped into 226 clusters using CD-HIT [25]. The C-terminal parts (starting from the last known N-terminal domain) of the 646 sequences without a known C-terminal domain were grouped into 301 clusters using CD-HIT. Twenty-five clusters contained five or more C-terminal parts and 38 clusters contained five or more N-terminal parts. Each of these clusters was subsequently used to construct a multiple sequence alignment with Clustal Omega [26] and a new HMM using HMMER (http://hmmer.org, accessed on 10 June 2022). Thus, in total, 63 new HMMs were constructed that were added to our set of Pfam domains to make final detections with. These new HMMs resulted in 601 additional detections (corresponding to an increase of 33.7% compared to the original number of detections).

### 3.3. Training an Extreme Gradient Boosting Classifier to Discriminate Phage RBPs from Other Phage Proteins

*RBP* and *Other* sequences of the training data were transformed into a feature representation using the ProtBert-BFD protein language model [28] and subsequently used to train a binary XGBoost classifier [27] to discriminate between both classes. Nested 4-fold cross-validation was used to both tune the hyperparameters in the inner loop and measure performance in the outer loop. The two hyperparameters that were tested were the maximum depth of each tree and the number of estimators. The optimal value for the maximum depth was three, while the optimal value for the number of boosting rounds turned out to be 500. The cross-validated classifier achieved an F1-score of 88.6% and a PR-AUC of 93.8%.

### 3.4. Benchmarking Our Parallel Approaches against PhANNs

Our domain-based and machine-learning-based approaches were benchmarked against the recently developed tool PhANNs [14] by making predictions for each of the protein sequences (*RBPs* and *Others*) in the held-out test set. Each method classified every test sequence either as an *RBP* or an *Other* sequence. All methods were compared by computing the F1-score, Matthews correlation coefficient, sensitivity, and specificity on the test set predictions (Table 3). Our XGBoost classifier resulted in the overall best performance, surpassing both PhANNs and the domain-based approach on almost all metrics (except for specificity that is the highest for the domain-based approach).

Finally, we constructed a Venn diagram for the RBP sequences in the test set, to illustrate the concordance and discordance among the different methods (Figure 5). The counts of the correct positive (RBP) predictions were visualized for each of the methods. Overall, the Venn diagram visually shows the superior performance of our XGBoost classifier, which was able to identify 1820 (1119 + 146 + 397 + 158) RBP sequences correctly (out of 1987). A total of 629 sequences were missed by the domain-based approach, indicating that many domains remain undetected even with our custom-developed set of HMMs. It also indicates the potential of machine-learning approaches to detect protein sequences with higher sensitivity compared to more traditional alignment-based approaches. In contrast, a total of 129 or 327 RBPs were missed by the XGBoost classifier and PhANNs, respectively, and 23 RBPs were missed by both methods (thus only detected with HMMs). This signals a (slight) gap in the knowledge of both models, leaving some room for further improvement of the models. Combining the protein embeddings with the scores of our HMM collection only resulted in a slight increase in performance across metrics on the held-out test set (Table 3), indicating that the protein embeddings by themselves already capture the information captured by the HMMs as well. Finally, 38 RBP sequences were missed by all three approaches.

## 4. Discussion

Although phage genome sequence data are becoming increasingly abundant, proper functional annotation remains scarce, is inconsistent, or even completely missing. Moreover, RBPs, despite having a conserved biological function, are specifically featured by a strong evolutionary diversification in response to the high diversity of host receptors, hampering traditional homology-based functional annotation. To alleviate this problem, we developed two parallel approaches to identify RBPs at the protein sequence level.

Our first approach consisted of collecting and constructing HMMs that represent protein domains strictly related to phage RBPs. This approach was inspired by the modularity of phage RBPs (typically consisting of an N-terminal structural domain combined with a C-terminal enzymatically active domain and/or binding domain) and the horizontal gene transfer events occurring across RBPs that result in this high modularity and diversity [18]. HMMs use a position-specific scoring system to capture information about the conservation and consensus of multiple sequence alignments, making them much more sensitive than typical pairwise methods such as BLAST [30]. A first drawback, however, is that RBP sequences cannot always clearly be divided into a structural (N-terminal) domain and receptor-binding or -cleaving (C-terminal) domain with the domains present in Pfam. Some Pfam domains cover only a portion of the N- or C-terminus, while in other cases multiple Pfam domains together constitute either the full structural or receptor-binding or -cleaving domain. There is currently no way around this drawback, although we can envision that structure predictions can be used to facilitate proper domain delineation and to construct HMMs corresponding to these domains in the future. The presence of additional chaperone, multimerization, or assembly domains at the C-terminus [31] further complicates the analysis. A second drawback is that few RBP-related protein domains are known in the Pfam database, necessitating the construction of custom HMMs from multiple sequence alignments to further increase the sensitivity of detections.

Our second approach focused on machine-learning models which can learn to detect complex patterns in data. We hypothesized that a machine-learning model, specifically trained to distinguish phage RBPs from other phage proteins, would also be able to detect RBPs at a higher sensitivity compared to traditional alignment-based methods. One drawback here is that both the language-based feature engineering and the gradient boosting modeling method result in a machine-learning approach that is not very interpretable. It is more straightforward to get a sense of why an RBP is detected based on a significant hit with a particular HMM than why an RBP is detected based on its particular protein language embedding that the XGBoost classifier recognizes as an RBP. However, both approaches could be combined to benefit from one another. Predictions made by the XGBoost model could be examined by our collection of HMMs. For those without a significant hit, new HMMs could be built and examined (e.g., by comparing to existing HMMs with HHsearch) [32]. This could not only validate the machine-learning predictions and make them more interpretable but, in turn, also lead to a more comprehensive collection of HMMs to make detections with. Indeed, various RBPs were correctly predicted by both PhANNs and our XGBoost model but were missed by the domain-based approach. This indicates that a (large) variety of annotated RBPs still consist of domains that were neither included in the manually collected HMMs from Pfam, nor were detected by the 63 HMMs that were custom-built afterwards. In that way, both approaches are complementary to one another.

Both approaches start from a comprehensive set of annotated RBP sequences that was carefully constructed based on the variety of keywords that are used to describe RBP sequences in public databases. This is a crucial first step in both building custom HMMs and machine-learning models that predict and generalize well. We showed that by focusing on a single class of proteins, we could implement a more indepth data collection approach, leading to superior model performance. As RBPs represent a crucially important class for practical applications, a tradeoff of having models predicting this single class of proteins versus predicting multiple classes is justified. Even though we attempted to make our data collection procedure as comprehensive as possible, we potentially still included sequences annotated as RBP while they are not. Sequences might be annotated with the wrong keyword and automatic annotation may cause new homologous sequences to be annotated wrongly as well. To avoid this bias as much as possible, our data processing included filtering steps for sequences with keywords such as ‘assembly’ and ‘hinge’ (among others) and deleted identical sequences across the *RBP* and *Other* classes. Overall, this bias is hard to overcome without sacrificing many likely correctly annotated sequences. One potential alternative solution would be to only consider experimentally determined sequences in UniProt, which would reduce the number of RBP sequences significantly, avoiding potential false positives, but likely also many true positives.

More generally, phage engineering efforts are becoming increasingly common. As phages often have a narrow host specificity, an alternative to performing a phage hunt is to change the host specificity of the phages that are already available. The underlying idea is that changing the host specificity by engineering phages may become faster compared to exhaustively looking for suitable phages. As the primary determinant of host specificity, RBPs are an important target to adjust the host specificity by modifying or swapping RBPs between phages [8,9,10]. In essence, the problem of finding suitable phages is then reduced to making suitable adjustments in their RBPs, given that no secondary specificity determinants (e.g., CRISPR immunity) interfere. However, modifying RBPs still often results in a loss of infectivity due to the complex positioning of the RBPs or loss of signal transmission [7]. The more RBP sequences we have available to analyze, the more we can learn from nature and understand the subtle differences that influence the host specificity. We strongly agree with the views of Lenneman et al. (2021), who claimed that high-throughput tools to identify phage RBPs, combined with structural information, can enable rapid engineering of phages at a sufficient scale for therapy [7]. More specifically, tools like AlphaFold and RoseTTAFold now allow for accurate predictions of three-dimensional protein structures of RBPs [33,34]. These structures will guide RBP engineering efforts in new ways.

The two approaches we have developed can serve as tools that tackle the first step towards realizing more effective RBP engineering efforts. By accurately identifying RBPs in phage genomes, we increase the pool of sequences that researchers can collect and investigate to guide host range adjustments. We provided open-source access to our code on GitHub (https://github.com/dimiboeckaerts/PhageRBPdetection, accessed on 10 June 2022) and Zenodo (https://doi.org/10.5281/zenodo.6491321, accessed on 10 June 2022) for researchers to replicate our analyses, construct their own RBP database based on the INPHARED phage genome collection, and identify new RBP sequences in genomic datasets. Jupyter notebooks (in Python) are available for researchers to use our HMMs to detect domains in protein sequences or compute embeddings for their sequences and pass them to our XGBoost models to make predictions.

## Figures and Tables

**Figure 1 viruses-14-01329-f001:**
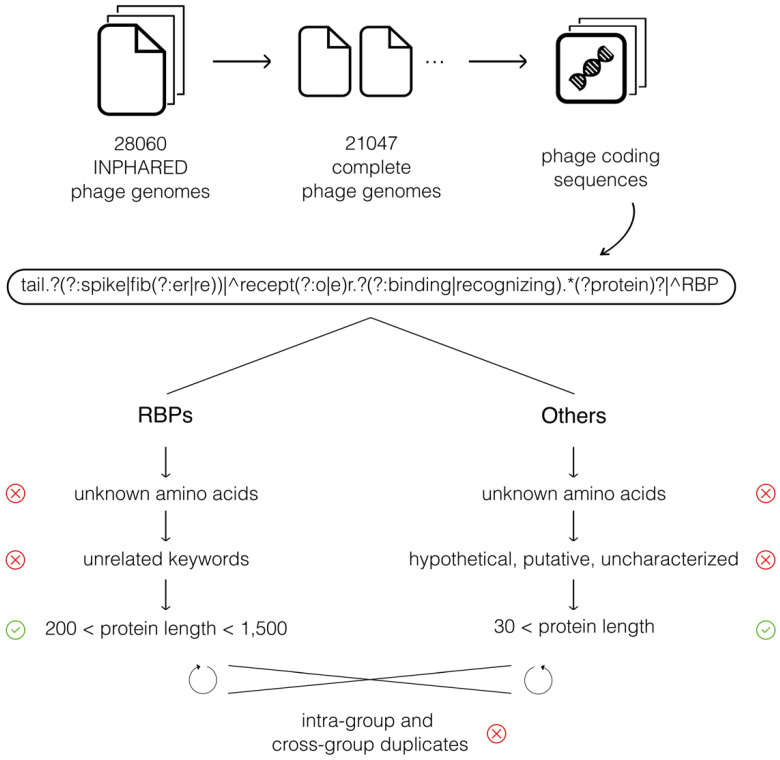
A schematic overview of data processing steps. Collected phage genomes from INPHARED were processed and coding sequences were extracted from each genome. Coding sequences were divided into two groups based on a regular expression that covered the various annotations of RBPs. Both groups were further processed to exclude sequences that had unknown amino acids, unwanted keywords, or extreme lengths.

**Figure 2 viruses-14-01329-f002:**
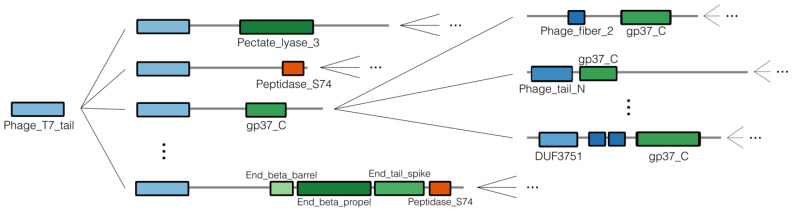
Visualization of the iterative manual search of RBP-related protein domains in the Pfam database, starting from the *Phage_T7_tail* domain. The iterative procedure entailed alternately identifying domains at the N- or C-terminus that were linked to the domains identified in the previous iteration of the manual search.

**Figure 3 viruses-14-01329-f003:**
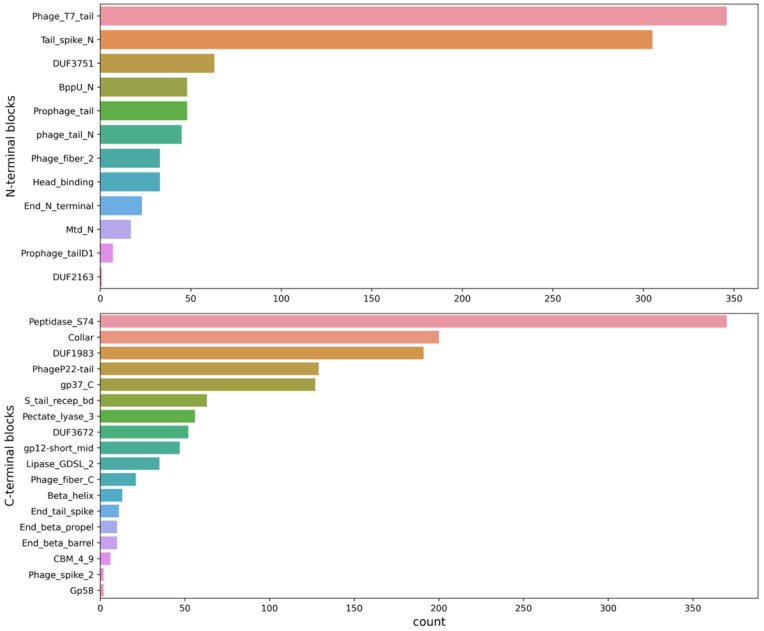
Count plot of the identified RBP-related HMMs from the Pfam database, grouped into N-terminal domains (**top**) and C-terminal domains (**bottom**). At the N-terminus, two domains occur substantially more than all the others (namely *Phage_T7_tail* and *Tail_spike_N*). At the C-terminus, *Peptidase_S74* is the most occurring domain.

**Figure 4 viruses-14-01329-f004:**
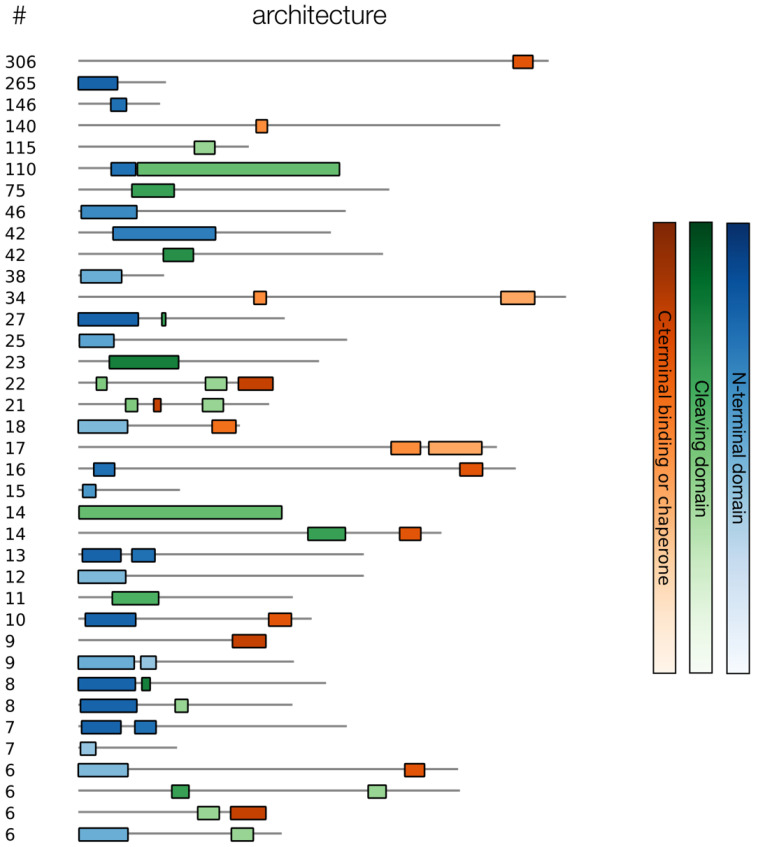
Architectures of the RBPs detected with RBP-related HMMs from the Pfam database, together with their number of occurrences and divided into three categories (N-terminal domains, cleaving domains, and C-terminal binding domain or chaperone). Examples of N-terminal domains are *Phage_T7_tail* and *Tail_spike_N*. Examples of cleaving domains are *Lipase_GDSL_2* and *Pectate_lyase_3*. Examples of binding domains and chaperones are *Phage_fiber_C* and *CBM_4_9*. Only the architectures that occurred more than five times were visualized. The top-five occurring architectures are single-domain architectures, indicating that there are many unknown RBP-related protein domains in the Pfam database.

**Figure 5 viruses-14-01329-f005:**
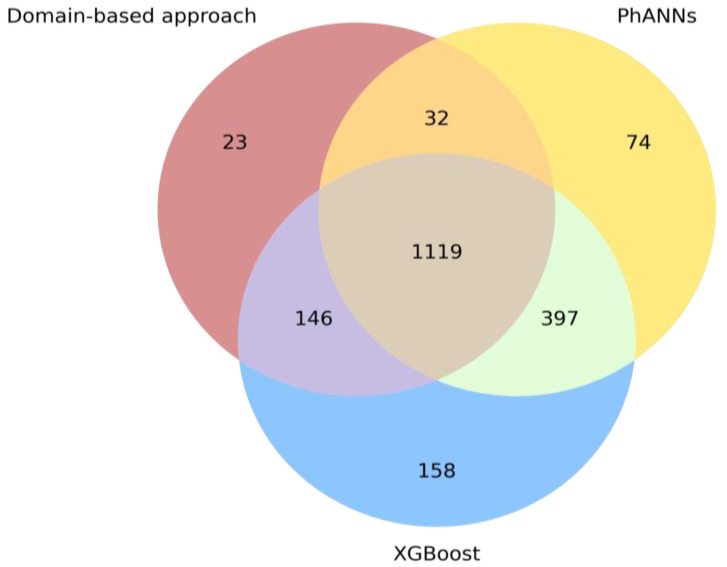
Venn diagram of the concordance between correct positive predictions made by the three benchmarked methods: our domain-based approach, our XGBoost classifier, and PhANNs.

**Table 1 viruses-14-01329-t001:** Annotation keywords for filtering RBPs.

**(a)** **Examples of Variations of Keywords That Pass or Fail the Constructed Regular Expression**	
Pass	tailspike, tail spike, tail-fiber, receptor-binding protein
Fail	spike, tail protein, binding protein
**(b)** **Additional Filtered Keywords after Applying the Regular Expression**	
** *RBPs* **	adaptor, wedge, baseplate, hinge, connector, structural, component, assembly, chaperone, attachment, capsid, proximal, measure
** *Others* **	probable, probably, uncharacterized, uncharacterised, putative, hypothetical, unknown, predicted

**Table 2 viruses-14-01329-t002:** RBP-related protein domains collected as HMMs from the Pfam database.

N-Terminal Domain	C-Terminal Domain
Phage_T7_tail	Lipase_GDSL_2
Tail_spike_N	Pectate_lyase_3
Prophage_tail	gp37_C
BppU_N	Beta_helix
Mtd_N	End_beta_propel
Head_binding	End_tail_spike
DUF3751	End_beta_barrel
End_N_terminal	PhageP22-tail
phage_tail_N	Phage_spike_2
Prophage_tailD1	gp12-short_mid
DUF2163	Collar
Phage_fiber_2	Peptidase_S74
	Phage_fiber_C
	S_tail_recep_bd
	CBM_4_9
	DUF1983
	DUF3672

**Table 3 viruses-14-01329-t003:** Benchmarked F1-scores, Matthews correlation coefficient (MCC) scores, sensitivity, and specificity for our domain-based approach, our XGBoost classifier, PhANNs, and an XGBoost with HMM scores combination on the held-out test data.

Method	F1-Score	MCC	Sensitivity	Specificity
Domain-based	72.0%	70.2%	66.4%	98.5%
PhANNs	69.8%	67.9%	81.6%	95.8%
XGBoost	84.0%	82.3%	91.6%	97.9%
XGBoost + HMM scores	84.8%	83.8%	92.2%	98.0%

## Data Availability

We provide open-source access to our code on GitHub (https://github.com/dimiboeckaerts/PhageRBPdetection, accessed on 1 June 2022) and through Zenodo (https://doi.org/10.5281/zenodo.6491321, accessed on 1 June 2022).

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
