# Peer review of "Identification of Phage Receptor-Binding Protein Sequences with Hidden Markov Models and an Extreme Gradient Boosting Classifier"

_viruses, 2022, doi:10.3390/v14061329_

Round 1
Reviewer 1 Report
Comments:
The manuscript submitted by Dimitri Boeckaerts et al. is concerned with developing two parallel approaches to alleviate the complex identification of RBP sequences in phage genomic data. Comparing to recently developed PhANNs tool, the methods reported here do show some advantages.
I support its publication. The following are some suggestions.
1. As shown in Fig.1, the candidate protein length is between 200-1500, but how to accurately define its N-terminus and C-terminus,why?
2. Line 110-111 “A full list of all discarded annotations is presented in Table 1b. Third, sequences shorter than 200 AAs or longer than 1500 AAs are discarded” vs. line 116 “sequences shorter than 30 AAs were removed”, confusing.
3. line124-125, pls also give the number of genome records for training OR for the benchmark against PhANNs.
4. Are the RBP sequences predicted by all the three approaches more convincing? any data supporting this point?If so, in practice, should we try all the three methods, and select the overlapping results?
5. Line342-348, currently, is there any approach dealing with this problem? if yes, pls also make a discussion.
Reviewer 2 Report
Boeckaerts et al., developed two approaches to identify new RBP sequences in phage genomic data. Although, the idea to change the specificity of bacteriophage by replacing its receptor-binding protein is very interesting and all methods to identify new RBPs attractive, I have some specific comments.
1. Please, add to the GitHub:
- all standalone versions of the program; (there are well-prepared notebooks, but subsequent steps are split into several separate parts, and this does not completely solve the case), Authors could make a script that only takes FASTA file (or possibly additional parameters that can be changed) and produces predictions.
lines 240 and 259-260 - all datasets (test, training; this is very important because, without it, it is impossible to verify whether the scores shown in the paper have been correctly calculated) and HMM with hits from them
2. Some links in the notebooks are not working:
for example in PhageRBPdetection/RBPdetect_notebook.ipynb
- Loading and processing data: getting MillardLab data and extracting annotated RBPs
- Custom HMM domains: constructing HMMs for enhanced RBP detection based on protein domains
- Machine learning models: constructing a XGBoost classifier to discriminate RBPs from nonRBPs.
- Benchmarking against PhANNs: comparing our methods against PhANNs on hold-out data.
3. line 100, 'Each CDS was checked for its annotation and assigned' – it is not clear how the Authors mapped the CDS sequences for their annotations. It was probably done at the INPHARED level, but it is not clearly stated in the text.
4. lines 110-115 - I am not sure if the cutoff < 30 aa for “Others” makes sense as it is very different than for RBP (200 – 1500 aa). It is justified from the biological point of view, but technically it can also lead to the development of a naive model that will learn that everything less than 200 is 'Other' and above 200 is RBP (indirectly this is related to the question of providing the training set and analyzing it, among others in terms of sequence length (e.g. histogram)).
5. lines 121-125 – Splitting both datasets into portions for training and testing according to date is not very fortunate. The data should be clustered (e.g. CD-HIT) and then randomly divided into training and testing (most likely the Authors' intention was to use the date as a guarantee that the method to which they referee (PhANN) was not able to use these sequences, but still this does not guarantee the safety of the procedure). Ideally, the Authors should cluster the data and sequences that are not similar to the ones used in the training set for PhANN should be used as a test set.
6. In the Results subsection 3.1 the number of sequences (positive and negative) for each set (RBPs and Others) was given but the problem is that the groups are not properly balanced (i.e. it would be much better to use the same number of RBP as 'Other' e.g. randomly excluding excess in 'Other').
7. Please, add to Table 3 other measures (AUC, MCC, Prec, Sens and TP, TN, FP, FN), additionally add to the supplement similar table, but for a testing set (it can be useful to assess the level of overfitting). In addition, there should be one more line in this table (Domain-based + XGBoost).
Round 2
Reviewer 2 Report
The authors satisfactorily answered all my queries.